# Reliability of Overground Running Measures from 2D Video Analyses in a Field Environment

**DOI:** 10.3390/sports7010008

**Published:** 2018-12-30

**Authors:** Lauralee Murray, C. Martyn Beaven, Kim Hébert-Losier

**Affiliations:** Adams Centre for High Performance, Faculty of Health, Sport and Human Performance, University of Waikato, Tauranga 3116, New Zealand; lauralee.murray@hotmail.co.nz (L.M.); martyn.beaven@waikato.ac.nz (C.M.B.)

**Keywords:** biomechanics, foot-strike, kinematics, running speed, test-retest

## Abstract

Two-dimensional running analyses are common in research and practice, and have been shown to be reliable when conducted on a treadmill. However, running is typically performed outdoors. Our aim was to determine the intra- and inter-rater reliability of two-dimensional analyses of overground running in an outdoor environment. Two raters independently evaluated 155 high-speed videos (240 Hz) of overground running from recreationally competitive runners on two occasions, seven days apart (test-retest study design). The reliability of foot-strike pattern (rear-foot, mid-foot, and fore-foot), foot-strike angle (°), and running speed (m/s) was assessed using weighted kappa (κ), percentage agreement, intraclass correlation coefficient (ICC), typical error (TE), and coefficient of variation (CV) statistics. Foot-strike pattern (agreement = 99.4%, κ = 0.96) and running speed (ICC = 0.98, TE = 0.09 m/s, CV = 2.1%) demonstrated excellent relative and absolute reliability. Foot-strike angle exhibited high relative reliability (ICC = 0.88), but suboptimal absolute reliability (TE = 2.5°, CV = 17.6%). Two-dimensional analyses of overground running outdoors were reliable for quantifying foot-strike pattern, foot-strike angle, and running speed, although foot-strike angle errors of 2.5° were typical. Foot-strike angle changes of less than 2.5° should be interpreted with caution in clinical settings, as they might simply reflect measurement errors.

## 1. Introduction

Running popularity is increasing, with over 5000 organized marathons and 2 million finishers per year since 2015, according to the Association of Road Racing Statisticians (http://www.arrs.net). The repetitive activation of the lower extremity muscles during running and the cyclical nature of this sporting activity has been linked to high injury rates [1,2], especially when combined with high vertical loading rates. Foot-strike pattern is an important part of running biomechanics, given that the foot provides a solid base of support [3], absorbs and redistributes impact forces throughout the kinetic chain, and also contributes to propulsion and balance during locomotion [1,3]. Foot-strike pattern in particular has been associated with an increased likelihood of certain types of running injuries [4]. For example, hip and knee injuries are two times more likely in rear-foot strikers than fore-foot strikers [4], with an increase in ankle and foot-related injuries observed in fore-foot strikers [4,5].

In a scientific and clinical context, only reproducible outcomes from testing procedures can be used to monitor small, but nonetheless functionally meaningful, changes in individuals. Despite the increasing scientific and clinical interest in foot-strike pattern and running gait retraining, there are relatively few studies investigating the reliability of foot-strike pattern [6,7,8,9]. These reliability studies have been conducted in a laboratory environment under controlled speed conditions. However, most running is performed outdoors. Although video analysis while running on a treadmill can provide valuable insight into overground running kinematics in a clinical context [10], the use of treadmills for overground running has been deemed impractical and not an accurate representation of an athlete’s habitual movement patterns in a field environment [11]. As such, it is unlikely that the reliability results from these prior studies [6,7,8,9] directly translate to real-world settings, especially given that the angle between the sole of the shoe and the ground at ground contact has been shown to be lower during treadmill running than overground running [12,13].

In laboratory and clinical settings, speed is generally determined and standardized using a treadmill [7,8]. The chosen assessment speed is either absolute (e.g., 3.2 m/s), relative (e.g., percentage of maximal), or self-selected, where self-selected can be based on habitual self-reported running speeds. When assessing running gait overground, runners are often required to target a selected speed where a margin of error of ±5% is deemed acceptable [14,15]; or similar to treadmill assessments, are asked to run at a self-selected speed [16]. Both of these approaches require the monitoring of speed with some form of equipment for proper documentation. Hand-held stop watches, photocells, global positioning systems, and laser-based timing devices are some of the most commonly used devices to monitor running speed in a field setting [17], and can provide quasi-instantaneous speed values. Two-dimensional (2D) video analyses are also frequently used, despite this method requiring further post-processing time for obtaining results. One common advantage across these field methods is their relative affordability compared to research-grade equipment. The standardization or monitoring of running speed is important in terms of the reproducibility of assessments and monitoring changes over time in runners.

The use of 2D video analyses in the field and during competitive events is common in sport science [18,19,20]. However, there is limited information on the reliability of measures of running in field-based settings. Indeed, the reliability of foot-strike pattern and angle measures are typically derived from treadmill-based analyses [6,7,8]. Within these settings, treadmills have been found to be overall reliable in terms of running gait analyses [7]. However, most runners train and compete outdoors, decreasing the validity and applicability of previous reliability studies for on-field assessments.

Given the common use of 2D video analyses within research and clinical practice to analyze running gait, and the importance of overground running assessments, the aims of this study were to determine the intra- and inter-rater reliability of 2D video analyses of overground running in an outdoor environment. In particular, we aimed to examine the reliability of foot-strike pattern, foot-strike angle, and running speed measures. Our hypothesis was that 2D video analysis of overground running performed outdoors is reliable for quantifying foot-strike pattern, foot-strike angle, and running speed. Determining both the intra- and inter-rater reliability of measures for this topic is important to ensure the appropriate interpretation of outcomes in clinical practice and science.

## 2. Materials and Methods

Twenty-eight recreational runners (17 males, 11 females) who were participating in a 12-km organized race volunteered to participate in this study (Table 1). Inclusion criteria were 18 years or over, free from any musculoskeletal or neurological injuries, and anticipated 12-km race times of 75 min or less (average race pace ≤ 6 min 15 s per km). Participants were recruited via electronic newsletters and emails sent by the race organizers, and on race day via pamphlets handed out at the registration desk and in the vicinity of the data collection area. All participants wore their own running shoes for testing and were asked to run at their perceived race pace during the running assessment. All participants provided written informed consent prior to participation. The protocol was pre-approved by the Human Research Ethics Committee of the University of Waikato (HREC(Health)#11) prior to recruitment of participants and complied with the Declaration of Helsinki.

The running gait of each participant was recorded pre- and post-race as part of a larger study investigating racing-induced fatigue. Data were collected on the same day, 5 to 60 min before runners started the race (i.e., pre-race) and 2 to 5 min after runners crossed the finish line (i.e., post-race). Given that intra- and inter-rater reliability of measures extracted from 2D videos was of interest here, the presence of fatigue in runners should not influence research outcomes.

Participants were asked to run three times at their perceived race pace (4.25 ± 0.71 m/s) through a 15-m level asphalt runway, with a 30-s walking rest between trials. Runners completed these trials both pre- and post-race, for a total of 6 running trials for each participant and 168 potentially eligible videos for intra- and inter-rater reliability assessment (28 participants × 2 sessions × 3 trials). The middle 5-m section of the runaway was demarcated by cones for video processing purposes. A digital camera (Cyber-shot DSC-RX10 II, Sony, Tokyo, Japan) with an actual focal length of 8.8 to 73.3 mm (35-mm equivalent focal length of 24–200 mm) sampling at 240 Hz was mounted on a 1-m high tripod to capture the sagittal plane of runners, 6 m away from the running area to the right-hand side of participants (i.e., camera at a 90° angle from the running area). Foot-strike pattern, foot-strike angle, and running speed were determined using the video recordings. Due to the on-field nature of the data collection, 13 of the potentially eligible videos were not available for subsequent reliability assessment (i.e., time constraints linked with the start of the 12-km organized race, operator error, and obscured participants from bystanders). Hence, reliability analyses were performed on 155 video recordings.

Siliconcoach Pro8 (The Tarn Group, Dunedin, N.Z.) was used to display each video recording frame by frame. The original video recordings were converted from MP4 to AVI format to ensure compatibility with the software. For each video, the foot-strike pattern and foot-strike angle for the right foot-strike occurrence nearest to the middle of the 15-m runway was determined (i.e., frame with the first clearly visible contact of the right foot with the ground) by each rater, independently. This particular foot-ground contact was selected because it was the one closest to the focal point of the camera and therefore less susceptible to camera-related distortional errors. Foot-strike pattern was classified from the 2D videos based on which part of the foot made ground contact as rear-foot (first contact was the heel or rear third of the sole only), mid-foot (first contact was the mid-foot or entire sole), or fore-foot (first contact was the fore-foot or front half of the sole), following previously reported classification schemes [18,19]. Foot-strike angle was calculated as the line that joined the sole of the shoe from the point of first contact and the horizontal plane of the running surface, wherein positive angles represented more pronounced rear-foot striking, and negative angles represented more pronounced fore-foot striking (Figure 1). The running speed of the participants was calculated based on the time taken for the right hip of the participants (i.e., the mid-portion of the pelvis) to cover the mid-5-m section of the runway. Data from our laboratory suggest that deriving running speed from Siliconcoach exhibits excellent concurrent validity (*r* = 0.98 [0.97, 0.98], coefficient of variation = 2.7% [2.5, 2.9], typical error = 0.07 m/s [0.07, 0.08], 90% confidence intervals [lower, upper]) against the Brower Timing Lights system (Brower Timing System, Draper, Utah, U.S.) using the same camera set-up.

To investigate the reliability of measures extracted (i.e., foot-strike pattern, foot-strike angle, and running speed), a repeated-measures design was employed. Data were extracted from all eligible videos (*n* = 155) by two sport science graduates (L.M., F.S.) on two separate occasions, 7 days apart. The two raters had more than 2 years of practical experience in strength and conditioning and practical assessment, and were accustomed to observing and quantifying human movement. Both raters completed the data extraction from the 155 eligible videos twice. Therefore, 310 comparisons were involved in determining intra-rater and inter-rater reliability values. Prior to data extraction, the raters familiarized themselves with the Siliconcoach Pro8 (The Tarn Group, Dunedin, N.Z.) software using the manufacturer’s online training resources. Furthermore, an internal data extraction protocol was developed and implemented in a series of internal training sessions to promote standardization. The two raters were blinded to each other’s measures, as well as to their previous measures when completing their second assessments. Intra-rater reliability was calculated by comparing Occasion 1 and Occasion 2 data from both raters, whereas inter-rater reliability was calculated by comparing Rater 1 and Rater 2 data from both occasions.

Mean and standard deviation (mean ± SD) values were computed to describe foot-strike angle and running speed data, whereas counts were used to describe foot-strike pattern data. Given that foot-strike pattern was a categorical variable with three levels (rear-foot, mid-foot, and fore-foot), linear weighted kappa (κ) with 90% confidence intervals [upper, lower] were computed to quantify the reliability. The agreement of the categorical ratings was interpreted as poor (κ < 0.40), fair (0.40 ≤ κ < 0.60), good (0.60 ≤ κ < 0.80), and excellent (κ ≥ 0.80) [6,21].

The reliability of foot-strike angle and speed data were analyzed using a customizable statistical spreadsheet [22]. Two-way mixed effects single measurement intraclass correlation coefficient (ICC [3,1]), typical error (TE), and coefficient of variation (CV) with 90% confidence intervals [lower, upper] were calculated to quantify the relative (ICC) and absolute (TE and CV) reliability of measures. For the purpose of interpreting the ICC, the relative reliability of measures was considered to be poor (ICC < 0.40), fair (0.40 ≤ ICC < 0.75), good (0.75 ≤ ICC < 0.90), and excellent (ICC ≥ 0.90) [23]. As is common practice in sport and exercise science [24], absolute reliability was deemed acceptable when the CV was < 10%, and suboptimal when the CV was ≥ 10%. Paired *t*-tests for intra-rater and independent *t*-tests with equal variance for inter-rater data were also carried out to identify significant changes in means between corresponding comparisons, with statistical significance set at *p* ≤ 0.05. 

## 3. Results

Based on the 155 videos analyzed, foot-strike pattern demonstrated excellent intra- and inter-rater reliability (Figure 2), with agreements of 99.4% [97.4, 99.9] and kappa values of 0.96 [0.92, 1.00]. While there was agreement in 99.4% of cases, there was disagreement on two of the videos analyzed, with one participant being classified rear-foot/mid-foot and another mid-foot/fore-foot. Intra- and inter-rater absolute and relative reliability was excellent for running speed (Table 2). Although relative reliability for foot-strike angle was good (ICC = 0.88), absolute reliability was suboptimal, with CV values of 17.6% (Table 2). A statistically significant difference in means was found between raters (*p* < 0.001) and occasions (*p* = 0.007) in terms of foot-strike angle measures, with one rater tending to rate higher than the other.

## 4. Discussion

The findings from this study were in agreement with our hypothesis and indicated that 2D video analysis of overground running performed outdoors was reliable for quantifying foot-strike pattern, foot-strike angle, and running speed, although foot-strike angle errors of 2.5° were typical within and between raters. As such, researchers and clinicians should interpret foot-strike angle changes of less than 2.5° with caution, as such changes might reflect the measurement error as opposed to an actual change in foot-strike pattern.

### 4.1. Foot-Strike Pattern

Foot-strike pattern is an important running characteristic, with research demonstrating differences between foot-strike patterns in vertical ground reaction forces [25], running biomechanics [25], and injury sites [26]. Our intra- and inter-rater reliability kappa values for foot-strike pattern classification (κ = 0.963) were higher than those previously reported from treadmill analyses [6,7,27]. Damsted, Larsen and Nielsen [7] reported kappa values for intra-rater agreement ranging from 0.63 to 0.69, and inter-rater agreement ranging from 0.41 to 0.53, whereas Pipkin, Kotecki, Hetzel and Heiderscheit [6] reported an average intra-rater and inter-rater kappa value of 0.86 and 0.85. Bertelsen, Jensen, Nielsen, Nielsen and Rasmussen [27] investigated inter-rater reliability of foot-strike classification of participants running on a laboratory runway, reporting kappa values for the left side of 0.76 to 0.82 and for the right side of 0.85 to 0.92. The lower kappa values reported in all three studies compared to ours could be due to the higher number of categories used to classify foot-strike pattern, with researchers using five (heel, heel/mid-foot, mid-foot, mid-foot/fore-foot, and fore-foot [7]) or four (heel, rear-foot, mid-foot, and fore-foot [6]; and rear-foot, mid-foot, fore-foot, and asymmetry [27]). Indeed, Damsted, Larsen and Nielsen [7] anticipated a lower reliability in foot-strike classification than previously reported [27] due to their use of five categories rather than the more typical three to four. However, these authors believed that their five-level classification had a greater clinical relevance, as their five-level classification considered extreme and subtle differences in foot-strike patterns [7]. The present study used a three-level foot-strike classification due to its greater ease of use and common application in practice and research [16,18,19,25,28]. Our results compared to the existing literature suggest that foot-strike classification is more reliable when using a lower number of categories, as recently indicated by Esculier, Silvini, Bouyer and Roy [9], who found 96.1% and 98.3% agreement between raters when considering three and two classification levels in runners with patellofemoral pain. Of the videos here analyzed, the raters only disagreed upon two occasions, with the disagreement spanning only one category (rear-foot/mid-foot and mid-foot/fore-foot). Closer inspection of the disagreement between raters revealed differences in the video frame identified as initial foot-ground contact, which would contribute to their disagreement in foot-strike classification.

The high level of agreement for foot-strike classification in our study compared to others might have resulted from our relatively homogenous sample, with 95% of videos being associated with a rear-foot strike as opposed to approximately 75% in previous reliability studies [6,8]. Each participant contributed between 3 to 6 videos to our reliability analysis, which promoted homogeneity. However, this homogeneity within the cohort was of a lesser concern given our interest in the rater reliability of measures. Furthermore, our higher proportion of rear-foot strikers was deemed to accurately reflect the recreationally competitive running population, where approximately 90% of individuals have been reported to be rear-foot strikers [18]. The high proportion of runners presenting with a rear-foot strike pattern might be a byproduct of running shoe characteristics, where more minimal shoes (i.e., lower mass, heel height, and heel-to-toe drop) are typically associated with a smaller foot-strike angle and fewer rear-foot foot strikes at initial ground contact compared to more cushioned shoes [29].

Running speed has also been shown to influence foot-strike pattern, with a lower proportion of rear-foot strikers at speeds of ≥5 m/s [30,31,32]. Thus, due to the average running speed of participants in the present study (~4.2 m/s), a greater proportion of rear-foot strikers was anticipated. Indeed, our findings agreed with existing literature that the recreational runner (albeit participating in organized racing events) generally adopts a rear-foot strike pattern, and they supported existing literature suggesting that runners running at speeds slower than 5 m/s are more likely to adopt a rear-foot strike pattern.

### 4.2. Foot-Strike Angle

The relative intra- and inter-rater reliability for foot-strike angle was good (ICC = 0.88), but the typical error of 2.5° was associated with a rather large CV (17.6%). The large CV here was likely a reflection of the foot-strike angle range in our population, which was limited to 42° [minimum value of −11° (fore-foot) and maximum of 31° (rear-foot)], which was similar to foot-strike angle ranges reported in other studies [31,33]. The foot-strike angle reliability measures derived herein can be useful in clinical and research settings to determine worthwhile changes in foot-strike angle. There is a growing amount of gait retraining literature attempting to influence foot-strike pattern [34]. Our study demonstrates that a change in foot-strike angle of at least 2.5° should be the minimum change required to infer an actual change in this measure, whereas a change of 2.5° or less would fall within the typical measurement error range. Similar to the foot-strike index proposed by Altman and Davis [33], we concur that the use of the foot-strike angle provides a more objective and quantifiable indicator of foot-strike pattern than using categorical variables.

Foot-strike angle measures for one rater and at the second occasion tended to be higher than those from the other rater and at the first rating occasion. This observation points to a difference between raters and occasions in terms of reference points used to calculate foot-strike angle, whereby placement of the line that joined the sole of the shoe from the point of first contact and the horizontal plane of the running surface slightly differed between raters and occasions. This difference was present despite steps taken to enhance standardization (i.e., development of a data extraction protocol and internal training sessions) and a one-week washout period between assessments. As such, caution is advised in interpreting differences between studies or combining foot-strike angle measures from various sources, as disparities in reference points have the potential to influence angular measures. 

### 4.3. Running Speed

Many running studies and clinical assessments of running gait use treadmills [6,7], which enables speed to be controlled and standardized across participants or testing occasions. Overground running speed is not as easy to standardize or quantify in the field, particularly when allowing individuals to self-select their running speed. A previous review of the literature suggested that video-based quantifications of speed are valid and reliable [17], with almost perfect agreement between speed computed from off-the-shelf video cameras and photocell timing systems [35], and no significant differences between 2D video analyses and laser measurement devices [36]. Excellent test-retest reliability for detecting the average speed of participants within a 3-m area using a video camera sampling at 50 and 100 Hz has been reported [36], with corresponding ICC values of 0.954 and 0.947. However, this particular reliability study examined the test-retest reliability of participant measures rather than the inter-rater reliability of video analyses [36]. Our research adds to the body of literature by identifying that both intra- and inter-rater reliability of running speed from 2D videos collected outdoors demonstrate excellent relative reliability (ICC = 0.98), with low typical error of measurements (0.09 m/s, ~2%).

### 4.4. Limitations

The findings from this study derive from a predominantly rear-foot striker population of runners, which might limit the generalization of findings to a sample of runners with a greater proportion of mid-foot and fore-foot strikers. Similarly, the reliability results stem from data extracted by two raters and two occasions separated by a period of seven days. The typical errors of 2.5° in foot-strike angle and 0.09 m/s in running speed found here provide reasonable estimates, although the magnitude of errors may differ with different raters and assessment timeframes. 

## 5. Conclusions

To summarize, the intra- and inter-rater reliability of foot-strike pattern identification during overground running using a high-speed video camera set-up in an outdoor environment demonstrated excellent reliability (κ = 0.96). Foot-strike angle and running speed using the same 2D video analysis also exhibited good to excellent relative reliability (ICC = 0.88 and 0.98, respectively), although errors of 2.5° were typical in foot-strike angle. Therefore, changes in foot-strike angles of less than 2.5° should be interpreted with caution in clinical and research settings, as they might simply reflect measurement errors as opposed to actual changes in foot-strike pattern.

## Figures and Tables

**Figure 1 sports-07-00008-f001:**
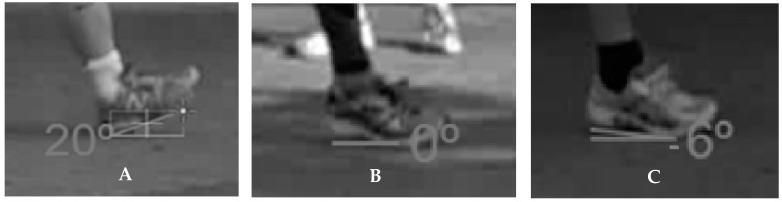
Foot-strike angle examples of rear-foot (left picture, **A**), mid-foot (middle picture, **B**), and fore-foot (right picture, **C**) strikes.

**Figure 2 sports-07-00008-f002:** Contingency tables reflecting the intra-rater (top tab, **A**) and inter-rater (bottom, **B**) agreement for foot-strike pattern classification.

**Table 1 sports-07-00008-t001:** Participant and shoe characteristics, mean ± standard deviation.

	Male (*n* = 17)	Female (*n* = 11)	Total (*n* = 28)
**Participants**			
Age (years)	37.8 ± 12.6	33.6 ± 10.0	36.2 ± 11.7
Height (cm)	176.5 ± 6.8	165.8 ± 6.9	172.1 ± 8.6
Body mass (kg)	81.1 ± 8.0	60.6 ± 6.5	73.6 ± 12.5
Running experience (years)	9.2 ± 10.3	5.4 ± 3.9	7.6 ± 8.3
Runs (per week)	3.9 ± 1.6	3.2 ± 0.6	3.6 ± 1.3
12-km race times (minutes)	58.9 ± 10.1	69.5 ± 12.0	63.0 ± 11.9
**Shoes**			
Mass (g)	306.7 ± 28.1	251.5 ± 35.1	284.6 ± 41.0
Heel height (mm)	28.3 ± 5.8	26.8 ± 6.2	27.7 ± 5.9
Heel-to-toe drop (mm)	9.8 ± 1.9	9.7 ± 1.3	9.8 ± 1.7

**Table 2 sports-07-00008-t002:** Mean ± standard deviation for foot-strike angle and speed for each occasion and rater. Typical error (TE), coefficient of variation (CV), intraclass coefficient (ICC) with 90% confidence intervals [lower, upper], and *p*-value statistics from intra-rater and inter-rater reliability analyses are also provided.

	Comparison ^1^		Statistics			
	1 (Raw Units)	2 (Raw Units)	TE (Raw Units)	CV (%)	ICC	*p*-Value
**Foot-Strike Angle (°)**						
Occasion 1 vs. 2	13.9 ± 7.1	14.5 ± 7.4	2.5 [2.3, 2.7]	17.6 [16.5, 18.8]	0.88 [0.86, 0.90]	0.007 *
Rater 1 vs. 2	15.2 ± 7.1	13.2 ± 7.4	2.5 [2.3, 2.7]	17.6 [16.5, 18.8]	0.88 [0.86, 0.90]	< 0.001 *
**Speed (m/s)**						
Occasion 1 vs. 2	4.25 ± 0.71	4.24 ± 0.71	0.09 [0.08, 0.09]	2.1 [2.0, 2.2]	0.98 [0.98, 0.99]	0.095
Rater 1 vs. 2	4.22 ± 0.70	4.28 ± 0.72	0.09 [0.08, 0.10]	2.1 [2.0, 2.3]	0.98 [0.98, 0.99]	0.276

^1^ Occasion 1 vs. 2: Intra-rater; Rater 1 vs. 2: Inter-rater. * Statistical significance (*p* < 0.05) from paired (intra) or independent (inter) *t*-tests with equal variance.

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
