# Peer review of "Reliability of Overground Running Measures from 2D Video Analyses in a Field Environment"

_sports, 2018, doi:10.3390/sports7010008_

Round 1

Reviewer 1 Report

The paper provides an data on the reliability of specific running data. Overall, the paper uses appropriate sample sizes and statistical analysis; it is also reasonably written. However, there are a number of areas where more information or clarity is required.

Introduction

1)     There seems to be a focus on running speed but the link between measurement changes associated with speed and reliability needs to be more explicit

2)     Line 42 to 43 – mention key variables but do not actually measure them – why not?

3)     Line 45 – I am unclear by what you mean by relative reliability in the context you have written it – please clarify

4)     At some point please can you verify the use of video captured speed. In the other examples provided (such as timing gates), speed can be given immediately and repeated trial used. How would speed knowledge via the collection of video be used by clinicians

5)     Line 55 replace on with in

6)     Why are intra and inter rater reliability important for this topic?

Method

1)      Why was pre-post data collected? Surely fatigue will have impacted the results? 

2)      how was the foot position relative to the camera standardized to avoid parallax and perspective errors

3)      How was running speed actually calculated? From the video it looks very closely zoomed so how could it be used to identify beginning and end of the running ‘zone’

4)      How do you know that it was a forefoot or heel landing? How can the data be used if in fact they were flat footed in reality but heel landing by observation.

5)      What type of ICC was used?

6)      You mention three trials pre-post and inter-rater. How were the trials used? Mean of the three? Why three? or were there an ICC between these individual trials?

7)      Line 133 – independent t-test is the term commonly used not unpaired t-test

Results

1)    Although agreement in identifying the landing, were the same participant videos identified as heel, flat or forefoot landing?

Discussion

1)      link to the participants and the limitations of this study that it is only related to these study participants. how may standard impact running variability? - how would the results translate from this population to others?

2)      Why was there a systematically higher score for the rater and occasion

3)      What is the impact of leg dominance on the results – you mention that Bertelsen, Jensen, Nielsen, Nielsen and Rasmussen looked at left and right.

4)      What is the impact of fatigue on the results (between occasions) particularly considering that they were significantly different?

5)      How would the running speed impact the results (i.e. small variations in speed along with individual differences in placement + angle measure errors due to camera-running alignments)

6)      Limitations and application of the results need discussion

Author Response

(Word document uploaded)

Manuscript ID: sports-394548
Title: Reliability of overground running measures from 2D video analyses in a field environment
Journal: Sports

Response: We appreciate the opportunity to submit a revised version of our manuscript to Sports for publication consideration. As requested, we have incorporated revisions according to the comments from your panel of expert reviewers and provide below a point-by-point summary of how we have addressed each comment. We have attempted to be as specific as possible in our responses. In addition, to facilitate the review process, we have indicated all modifications in the manuscript in RED font colour, except for removed text where there is no colour coding. We would like to thank the reviewers for their critical appraisal of and feedback on our manuscript. The reviewers were very thorough and we feel that the advice given has enhanced the impact and quality of our manuscript.

Reviewer 1

The paper provides an data on the reliability of specific running data. Overall, the paper uses appropriate sample sizes and statistical analysis; it is also reasonably written. However, there are a number of areas where more information or clarity is required.

Response: We thank the reviewer for the overall appreciation of our work and appreciate the constructive criticism provided.

Introduction

1) There seems to be a focus on running speed but the link between measurement changes associated with speed and reliability needs to be more explicit

Response: We agree with the reviewer that our introduction was overly focused on running speed and changes in running with speed. A more ‘reliability-focused’ introduction is more suited to the aims of the paper and content of the manuscript. As such, we have replaced our second paragraph with one that focuses on the scientific literature addressing the reliability of foot strike pattern.

Introduction: In a scientific and clinical context, only reproducible outcomes from testing procedures can be used to monitor small, but nonetheless functionally meaningful, changes in individuals. Given the increasing scientific and clinical interest in foot-strike pattern and running gait retraining, there are relatively few studies investigating the reliability of foot-strike pattern [6-9]. These reliability studies have been conducted in a laboratory environment under controlled speed conditions; however, most running is performed outdoors. Although video analysis while running on a treadmill can provide valuable insight into overground running kinematics in a clinical context [10], the use of treadmills for overground running has been deemed impractical and not an accurate representation of athlete’s habitual movement patterns in a field environment [11]. As such, it is unlikely the reliability results from these prior studies[6-9] directly translate to real-world settings, especially given that the angle between the sole of the shoe and ground at ground contact has been shown to be lower during treadmill than overground running [12,13].

2) Line 42 to 43 – mention key variables but do not actually measure them – why not?

Response: With the replacement of paragraph 2, these lines no longer appear in our manuscript.

3)     Line 45 – I am unclear by what you mean by relative reliability in the context you have written it – please clarify

Response: We have added an example in parentheses.

Introduction: The chosen assessment speed is either absolute (e.g., 3.2 m/s), relative (e.g., percentage of maximal), or self-selected, where self-selected can be based on habitual self-reported running speeds.

4)     At some point please can you verify the use of video captured speed. In the other examples provided (such as timing gates), speed can be given immediately and repeated trial used. How would speed knowledge via the collection of video be used by clinicians

Response: Indeed, the use of 2D videos to monitor speed in a field environment still requires post-processing time. We now more clearly make this distinction between field measures clear.

Introduction: Hand-held stop watches, photocells, global positioning systems, and laser-based timing devices are some of the most commonly used devices to monitor running speed in a field setting [14], which can provide quasi-instantaneous speed values. Two dimensional (2D) video analyses are also frequently used despite this method requiring further post-processing time for obtaining results. One common advantage across these methods is their relative affordability compared to research-grade equipment.

5)     Line 55 replace on with in

Response: Change applied.

6)     Why are intra and inter rater reliability important for this topic?

Response: In a scientific and clinical context, only reproducible outcomes from testing procedures can be used to monitor small, but nonetheless functionally meaningful, changes in individuals. Determining both the intra- and inter-rater reliability of measures for this topic is important to ensure the appropriate interpretation of outcomes in clinical practice and science. These justifications have been added to our Introduction in Paragraph 2 and Paragraph 5, respectively.

Introduction: In a scientific and clinical context, only reproducible outcomes from testing procedures can be used to monitor small, but nonetheless functionally meaningful, changes in individuals.

Introduction: Determining both the intra- and inter-rater reliability of measures for this topic is important to ensure the appropriate interpretation of outcomes in clinical practice and science.

Method

1)      Why was pre-post data collected? Surely fatigue will have impacted the results?

Response: Pre and post data were collected as part of a larger study investigating racing-induced fatigue. Fatigue in runners should not impact the intra- and inter-rater reliability of results, as the reliability of the video analysis was of interest (not of the reliability of the runners). If anything, the increase heterogeneity should improve the data set. The following statements have been added to justify the pre-post data collection, and specify that the reliability results should not be impacted.

Methods: The running gait of each participant was recorded pre- and post-race as part of a larger study investigating racing-induced fatigue. Given that intra- and inter-rater reliability of measures extracted from 2D videos was of interest here, the presence of fatigue in runners should not influence outcomes.

2)      how was the foot position relative to the camera standardized to avoid parallax and perspective errors

Response: The camera was mounted to capture the sagittal plane of runners to the right-hand side of participants (i.e., camera at a 90° angle from the running area). For each video, the foot-strike pattern and foot-strike angle for the right foot-strike occurrence nearest to the middle of the 15-m runway was determined (i.e., frame with the first clearly visible contact of the right foot with the ground).  This particular foot-ground contact was selected as it was the one closest to the focal point of the camera and therefore less susceptible to camera-related distortional errors.

The following statements have been added to the methods section to specify further.

Methods: …(i.e., camera at a 90° angle from the running area)

Methods: This particular foot-ground contact was selected as it was the one closest to the focal point of the camera and therefore less susceptible to camera-related distortional errors.

3)      How was running speed actually calculated? From the video it looks very closely zoomed so how could it be used to identify beginning and end of the running ‘zone’

Response: The zoomed-in pictures were inserted to the manuscript to highlight how foot-strike angle was calculated. Running speed of participants was calculated based on the time taken for the right hip of participants (i.e., mid-portion of the pelvis) to cover the mid 5-m section of the runway. The following statement was added to the methods section.

Methods: Running speed of participants was calculated based on the time taken for the right hip of participants (i.e., mid-portion of the pelvis) to cover the mid 5-m section of the runway.

4)      How do you know that it was a forefoot or heel landing? How can the data be used if in fact they were flat footed in reality but heel landing by observation.

Response: As noted in the methods section, foot-strike pattern was classified based on which part of the foot made initial ground contact as: rear-foot (first contact was the heel or rear third of the sole only), mid-foot (first contact was the mid-foot or entire sole), or fore-foot (first contact was the fore-foot or front half of the sole) following previously reported classification schemes. These classification schemes are employed to determine foot-strike pattern from 2D videos. We have now added the fact that foot strike pattern was classified from the 2D videos.

Methods: Foot-strike pattern was classified from the 2D videos based on (…)

5)      What type of ICC was used?

Response: A two-way mixed effects single measurement ICC(3,1) was used. We have added this specification to our manuscript.

Methods: A two-way mixed effects single measurement ICC(3,1)…

6)      You mention three trials pre-post and inter-rater. How were the trials used? Mean of the three? Why three? or were there an ICC between these individual trials?

Response: Each trial provided one video that was used to determine the intra-rater and inter-rater reliability of 2D video analyses. Hence, each participant there was 168 potentially eligible videos for intra- and inter-rater reliability assessment (28 participants x 2 sessions x 3 trials). Due to the on-field nature of the data collection, 13 of the potentially eligible videos were not available for subsequent reliability assessment, and 155 videos were employed. This information is specified in the methods section. Three trials were collected pre and post as this is the number of trials typically used in a research context.

7)      Line 133 – independent t-test is the term commonly used not unpaired t-test

Response: Change applied.

Results

1)    Although agreement in identifying the landing, were the same participant videos identified as heel, flat or forefoot landing?

Response: There was disagreement on two of the videos analysed, with one participant being classified rear-foot / mid-foot and another mid-foot / fore-foot. This specification has been added to the results section

Results: While there was agreement in 99.4% of cases, there was disagreement on two of the videos analysed, with one participant being classified rear-foot / mid-foot and another mid-foot / fore-foot.

Discussion

1)      link to the participants and the limitations of this study that it is only related to these study participants. how may standard impact running variability? - how would the results translate from this population to others?

Response: We thank the reviewer for the comments. We have added a section that addresses the limitations of our research study.

2)      Why was there a systematically higher score for the rater and occasion

Response: This observation points to a difference between raters and occasions in terms of reference points used to calculate foot-strike angle. We have added a section in our Discussion to address the systematically higher scores for one rater

Discussion: Foot-strike angle measures for one rater and at the second occasion tended to be higher than those from the other rater and at the first rating occasion. This observation points to a difference between raters and occasions in terms of reference points used to calculate foot-strike angle, whereby placement of the line that joined the sole of the shoe from the point of first contact and the horizontal plane of the running surface slightly differed between raters. This difference was present despite steps taken to enhance standardisation (i.e., development of a data extraction protocol and internal training sessions) and a 1-week washout period between assessments. As such, caution is advised in interpreting differences between studies or combining foot-strike angle measures from various sources as disparities in reference points has the potential to influence angular measures.

3)      What is the impact of leg dominance on the results – you mention that Bertelsen, Jensen, Nielsen, Nielsen and Rasmussen looked at left and right.

Response: Leg dominance of runners should not impact the intra- and inter-rater reliability of results, as the reliability of the video analysis was of interest (not the between-leg difference of the runners).

4)      What is the impact of fatigue on the results (between occasions) particularly considering that they were significantly different?

Response: As noted above, fatigue in runners should not impact the intra- and inter-rater reliability of results, as the reliability of the video analysis was of interest (not of the reliability of the runners). If anything, the increase heterogeneity should improve the data set. The following statements have been added to justify the pre-post data collection, and specify that the reliability results should not be impacted.

Methods: The running gait of each participant was recorded pre- and post-race as part of a larger study investigating racing-induced fatigue. Given that intra- and inter-rater reliability of measures extracted from 2D videos was of interest here, the presence of fatigue in runners should not influence outcomes.

5)      How would the running speed impact the results (i.e. small variations in speed along with individual differences in placement + angle measure errors due to camera-running alignments)

Response: We thank the reviewer for the comments. Running speed should not impact the intra- and inter-rater reliability of results, as the reliability of the video analysis was of interest (not the effect of speed on foot-strike pattern or angle).

6)      Limitations and application of the results need discussion

Response: We thank the reviewer for the comments. We have added a section that addresses the limitations of our research study.

Reviewer 2 Report

The manuscript is well-written and the general topic is one of current interest in the running community.  The specific topic is not particularly novel but contributes to the current body of literature in the specific nature of the test population and setting. The writing style was clear and there was a good level of detail in most of the manuscript. Statistical analyses were well-designed and clearly reported. Comments follow about the introduction and the raters to explain the "can be improved" ratings provided above.

Comments:

I found the introduction interesting but I was confused by the focus on running speed and foot-strike pattern for this particular study. It would make more sense to focus on what has been done with gait analysis using video analysis. Discussion of studies using 2D video to assess any kinematic measures, even during walking, would be more relevant. A strength of the study was that it was set up at a racing location on race day - perhaps lead up to that point more.

line 81: can the authors add any descriptors regarding how long before or after the race the tests were performed? At minimum, note that it was the same day?

Inter-rater reliability design can be improved: only two raters with the same training were compared in this study. Ref [24] also only had 2 raters, but their conclusions were not about reliability; Ref [11] and 18] had 2 and 3 raters who were experienced therapists, presumably with different training. So the design choice of this study is not far from what has been down in the literature. However, it is a limitation and, at minimum, the discussion regarding 2.5deg difference in foot angle should include a comment related to this design decision.

To clarify in methods: did each rater perform the steps described in lines 95-105? If the exact time of contact was not provided to the raters, then the foot strike pattern and angle measures would be dependent on two subjective measures - time of contact and classification/measurement of the angle. The authors should recognize this point and clarify. (note: reliability comparisons could also be done on time of contact)

Line 105: were specific instructions used to determine when the runner covered the 5msection of runway?  ex: when a particular body part passed the first and last cones?

Shoe descriptions in Table 1 are interesting.  Perhaps comments regarding their effect on foot angle could be included in discussion?

Table 2.  Order of measures in the caption should match the order of the measures in the table.

Some suggested References. The following references are relevant and might add to the paper.

Esculier et al. "Video-based assessment of foot strike pattern and step-rate...".  Physical Therapy in Sport (2018)

Fellin et al. "Comparison of methods for kinematic identification of footstrike and toe-off during overground and treadmill running" (2010)

Hanley & Mohan "Changes in gait during constant pace treadmill running" (2014)

Author Response

Manuscript ID: sports-394548
Title: Reliability of overground running measures from 2D video analyses in a field environment
Journal: Sports

Response: We appreciate the opportunity to submit a revised version of our manuscript to Sports for publication consideration. As requested, we have incorporated revisions according to the comments from your panel of expert reviewers and provide below a point-by-point summary of how we have addressed each comment. We have attempted to be as specific as possible in our responses. In addition, to facilitate the review process, we have indicated all modifications in the manuscript in RED font colour, except for removed text where there is no colour coding. We would like to thank the reviewers for their critical appraisal of and feedback on our manuscript. The reviewers were very thorough and we feel that the advice given has enhanced the impact and quality of our manuscript.

Reviewer 2

The manuscript is well-written and the general topic is one of current interest in the running community.  The specific topic is not particularly novel but contributes to the current body of literature in the specific nature of the test population and setting.

The writing style was clear and there was a good level of detail in most of the manuscript. Statistical analyses were well-designed and clearly reported. Comments follow about the introduction and the raters to explain the "can be improved" ratings provided above.

Response: We thank the reviewer for the overall appreciation of the detail, writing style, and statistical analyses of our work and appreciate the constructive criticism provided.

Comments:

I found the introduction interesting but I was confused by the focus on running speed and foot-strike pattern for this particular study. It would make more sense to focus on what has been done with gait analysis using video analysis. Discussion of studies using 2D video to assess any kinematic measures, even during walking, would be more relevant. A strength of the study was that it was set up at a racing location on race day - perhaps lead up to that point more.

Response: We agree with the reviewer that our introduction was overly focused on running speed and changes in running with speed. As noted above, a more ‘reliability-focused’ introduction is more suited to the current paper and, as such, we have replaced our second paragraph.

Introduction: In a scientific and clinical context, only reproducible outcomes from testing procedures can be used to monitor small, but nonetheless functionally meaningful, changes in individuals. Given the increasing scientific and clinical interest in foot-strike pattern and running gait retraining, there are relatively few studies investigating the reliability of foot-strike pattern [6-9]. These reliability studies have been conducted in a laboratory environment under controlled speed conditions; however, most running is performed outdoors. Although video analysis while running on a treadmill can provide valuable insight into overground running kinematics in a clinical context [10], the use of treadmills for overground running has been deemed impractical and not an accurate representation of athlete’s habitual movement patterns in a field environment [11]. As such, it is unlikely the reliability results from these prior studies[6-9] directly translate to real-world settings, especially given that the angle between the sole of the shoe and ground at ground contact has been shown to be lower during treadmill than overground running [12,13].

line 81: can the authors add any descriptors regarding how long before or after the race the tests were performed? At minimum, note that it was the same day?

Response: We now specify that the timeframe of the pre (between 5 to 90 minutes before the race) and post (between 2 to 5 minutes after the race) data collection sessions. These sessions were indeed conducted on the same day.

Methods: Data were collected on the same day, 5 to 60 minutes before runners started the race (i.e., pre-race) and 2 to 5 minutes after runners crossed the finish line (i.e., post-race).

Inter-rater reliability design can be improved: only two raters with the same training were compared in this study. Ref [24] also only had 2 raters, but their conclusions were not about reliability; Ref [11] and 18] had 2 and 3 raters who were experienced therapists, presumably with different training. So the design choice of this study is not far from what has been down in the literature. However, it is a limitation and, at minimum, the discussion regarding 2.5deg difference in foot angle should include a comment related to this design decision.

Response: We thank the reviewer for the comment, and agree that investigating reliability from three raters and occasions would provide more information than data from two raters and two occasions. The typical errors determined in our study provide an estimate, and might change with different raters or assessment timeframes. We agree that discussing this point is of relevance, and have added the following to our limitation section.

Limitations: Similarly, the reliability results stem from data extracted by two raters and two occasions separated by a period of 7 days. The typical errors of 2.5° in foot-strike angle and 0.09 m/s in running speed found here provide reasonable estimates, although the magnitude of errors may differ with different raters and assessment timeframes.

To clarify in methods: did each rater perform the steps described in lines 95-105? If the exact time of contact was not provided to the raters, then the foot strike pattern and angle measures would be dependent on two subjective measures - time of contact and classification/measurement of the angle. The authors should recognize this point and clarify. (note: reliability comparisons could also be done on time of contact)

Response: Yes, the steps enumerated in the methods section were performed by each rater independently from each other (now clarified). We agree that the difference in time of contact identification can contribute to the difference in foot strike pattern classification and angles (i.e., previously addressed in lines 183 – 185). A reliability comparison on these time of contacts could have been performed; however, this particular discrete measure was not of particular interest given that the exact time frame of foot contact indentification alone has no practical relevance (as opposed to foot strike angle, foot strike pattern, and running speed).

Methods: For each video, the foot-strike pattern and foot-strike angle for the right foot-strike occurrence nearest to the middle of the 15-m runway was determined (i.e., frame with the first clearly visible contact of the right foot with the ground) by each rater, independently.

Line 105: were specific instructions used to determine when the runner covered the 5m section of runway?  ex: when a particular body part passed the first and last cones?

Response: Running speed of participants was calculated based on the time taken for the right hip of participants (i.e., mid-portion of the pelvis) to cover the mid 5-m section of the runway. The following statement was added to the methods section.

Methods: Running speed of participants was calculated based on the time taken for the right hip of participants (i.e., mid-portion of the pelvis) to cover the mid 5-m section of the runway.

Shoe descriptions in Table 1 are interesting.  Perhaps comments regarding their effect on foot angle could be included in discussion?

Response: We thank the reviewer for noticing, and agree that this is of interest. We have added a comment relating to shoe characteristics in our discussion section.

Discussion: The high proportion of runners presenting with a rear-foot strike pattern might be a by-product of running shoe characteristics, whereby more minimal shoes (i.e., lower mass, heel height, and heel-to-toe drop) are typically associated with a smaller foot-strike angle and less rear-foot at initial ground contact compared to more cushioned shoes [26].

Table 2.  Order of measures in the caption should match the order of the measures in the table.

Response: We have changed the caption for Table 2 to reflect the order of measures.

Table 2. Mean ± standard deviation for foot-strike angle and speed for each rater and occasion are provided. Typical error (TE), coefficient of variation (CV), and intraclass coefficient (ICC) with 90% confidence intervals [lower, upper] and p-value statistics from intra-rater and inter-rater reliability analyses are also provided.

Some suggested References. The following references are relevant and might add to the paper.

Esculier et al. "Video-based assessment of foot strike pattern and step-rate...".  Physical Therapy in Sport (2018)

Fellin et al. "Comparison of methods for kinematic identification of footstrike and toe-off during overground and treadmill running" (2010)

Hanley & Mohan "Changes in gait during constant pace treadmill running" (2014)

Response: We thank the reviewer for the suggestions. The paper from Esculier et al. is cited in our discussion section. We have added a sentence to refer to Hanley & Mohan in our introduction. The paper of Fellin et al. was not integrated as it compares methods for the kinematic identification of foot-strike and toe-off using 3D methods rather than 2D methods.

Introduction: Although video analysis while running on a treadmill can provide valuable insight into overground running kinematics in a clinical context [10], the use of treadmills for overground running has been deemed impractical and not an accurate representation of athlete’s habitual movement patterns in a field environment [11].

Discussion (pre revision): Our results compared to existing literature suggest that foot-strike classification is more reliable when using a lower number of categories, as recently indicated by Esculier, et al. [26] who found 96.1 and 98.3% agreement between raters when considering three and two classification levels in runners with patellofemoral pain.

Reviewer 3 Report

The paper is interesting, well written and the results are convincing. However, the Reviewer recommends considering the comments listed below.

The Reviewer draw the Author's attention on two further publications related to the vide analysis of the kinematics of running. Paper [1] discusses many ideas about 2D video analyses. In [2], the measurement of the speed, the stride length etc. is also based on 2D video analysis for overground running. These studies could be also mentioned in the manuscript.

The issue of overground and treadmill running is discussed in [3]. The reviewer suggest considering the citation of this article too and include some more explanation (in the 4th paragraph of the Introduction Section) of the possible differences between treadmill and overground running measurements.

The second paragraph of Section Materials and Methods explains the technical details of the camera setup. The first issue here: what was the focal length of the camera? Please clarify it. The second issue is that the 6m distance between the camera and the running track seems to be quite small comparing to the 5m long section. The reviewer's concern is that the smaller the camera-object distance, the larger the optical distortion. The Authors should explain why the optical distortion does not have an effect on the results.

The measurement of the foot strike angle is explained in the 3rd paragraph of Section Materials and Methods. It may be better understandable, if the difference between the angle of the shoe sole and the angle of the foot would be more emphasized in the text.

The measurement of the running speed is explained in the 3rd paragraph of Section Materials and Methods. It is not clear which point of the body was used as a reference. How do the authors mean that covering the 5m section of the runway? A specific point of the trunk or the head has to pass the 5m long section? Please clarify it in the text.

The last paragraph of Section Materials and Methods includes an URL citation, which should appear in the list of references.

[1] Souza, R.B. : ''An Evidence-Based Videotaped Running Biomechanics Analysis''. Physical Medicine and Rehabilitation Clinics of North America, 2016, vol.27, no.1, pp217-36. DOI:10.1016/j.pmr.2015.08.006.

[2] Bencsik, L. and Zelei, A.: ''Effects of human running cadence and experimental validation of the bouncing ball model''. Mechanical Systems and Signal Processing, 2016, vol.89, pp78-87. DOI:10.1016/j.ymssp.2016.08.001.

[3] Fellin, R.E. and Manal, K and Davis, I.S.: ''Comparison of Lower Extremity Kinematic Curves During Overground and Treadmill Running''. Journal of Applied Biomechanics, 2010, vol.26, no.4, pp407-414.

Author Response

Manuscript ID: sports-394548
Title: Reliability of overground running measures from 2D video analyses in a field environment
Journal: Sports

Response: We appreciate the opportunity to submit a revised version of our manuscript to Sports for publication consideration. As requested, we have incorporated revisions according to the comments from your panel of expert reviewers and provide below a point-by-point summary of how we have addressed each comment. We have attempted to be as specific as possible in our responses. In addition, to facilitate the review process, we have indicated all modifications in the manuscript in RED font colour, except for removed text where there is no colour coding. We would like to thank the reviewers for their critical appraisal of and feedback on our manuscript. The reviewers were very thorough and we feel that the advice given has enhanced the impact and quality of our manuscript.

Reviewer 3

The paper is interesting, well written and the results are convincing. However, the Reviewer recommends considering the comments listed below.

Response: We thank the reviewer for the overall appreciation of our work written work and results and for the constructive criticism provided.

The Reviewer draw the Author's attention on two further publications related to the vide analysis of the kinematics of running. Paper [1] discusses many ideas about 2D video analyses. In [2], the measurement of the speed, the stride length etc. is also based on 2D video analysis for overground running. These studies could be also mentioned in the manuscript.

The issue of overground and treadmill running is discussed in [3]. The reviewer suggest considering the citation of this article too and include some more explanation (in the 4th paragraph of the Introduction Section) of the possible differences between treadmill and overground running measurements.

Response: We thank the reviewer for the papers suggested. We have integrated paper [1] and paper [3] to our manuscript, which has helped to highlight possible differences between treadmill and overground running measurements.

Introduction: In a scientific and clinical context, only reproducible outcomes from testing procedures can be used to monitor small, but nonetheless functionally meaningful, changes in individuals. Given the increasing scientific and clinical interest in foot-strike pattern and running gait retraining, there are relatively few studies investigating the reliability of foot-strike pattern [6-9]. These reliability studies have been conducted in a laboratory environment under controlled speed conditions; however, most running is performed outdoors. Although video analysis while running on a treadmill can provide valuable insight into overground running kinematics in a clinical context [10], the use of treadmills for overground running has been deemed impractical and not an accurate representation of athlete’s habitual movement patterns in a field environment [11]. As such, it is unlikely the reliability results from these prior studies[6-9] directly translate to real-world settings, especially given that the angle between the sole of the shoe and ground at ground contact has been shown to be lower during treadmill than overground running [12,13].

The second paragraph of Section Materials and Methods explains the technical details of the camera setup. The first issue here: what was the focal length of the camera? Please clarify it. The second issue is that the 6m distance between the camera and the running track seems to be quite small comparing to the 5m long section. The reviewer's concern is that the smaller the camera-object distance, the larger the optical distortion. The Authors should explain why the optical distortion does not have an effect on the results.

Response: We thank the reviewer for the comments. We now specify the focal length of the camera, and address the potential for distortional errors. Data from our laboratory suggest a limited effect of optical distortion on speed values, with running speed from Siliconcoach exhibiting excellent concurrent validity (r = 0.98 [0.97, 0.98], coefficient of variation = 2.7% [2.5, 2.9], typical error = 0.07 m/s [0.07, 0.08], 90% confidence intervals [lower, upper]) against the Brower Timing Lights system (Brower Timing System, Colorado, USA) using the same camera set-up. This last information has also been added to our methods section.

Methods: A digital camera (Cyber-shot DSC-RX10 II, Sony, Tokyo, Japan) with an actual focal length of 8.8 to 73.3 mm (35 mm equivalent focal length of 24 – 200 mm)…

Methods: This particular foot-ground contact was selected as it was the one closest to the focal point of the camera and therefore less susceptible to camera-related distortional errors.

Methods: Data from our laboratory suggests that deriving running speed from Siliconcoach exhibits excellent concurrent validity (r = 0.98 [0.97, 0.98], coefficient of variation = 2.7% [2.5, 2.9], typical error = 0.07 m/s [0.07, 0.08], 90% confidence intervals [lower, upper]) against the Brower Timing Lights system (Brower Timing System, Colorado, USA) using the same camera set-up.

The measurement of the foot strike angle is explained in the 3rd paragraph of Section Materials and Methods. It may be better understandable, if the difference between the angle of the shoe sole and the angle of the foot would be more emphasized in the text.

Response: We trust that the description and Figure 1 provide a sufficient description of how foot strike angle was measured.

Methods: Foot-strike angle was calculated as the line that joined the sole of the shoe from the point of first contact and the horizontal plane of the running surface, wherein positive angles represent more pronounced rear-foot striking, and negative angles represent more pronounced fore-foot striking (Figure 1).

The measurement of the running speed is explained in the 3rd paragraph of Section Materials and Methods. It is not clear which point of the body was used as a reference. How do the authors mean that covering the 5m section of the runway? A specific point of the trunk or the head has to pass the 5m long section? Please clarify it in the text.

Response: Running speed of participants was calculated based on the time taken for the right hip of participants (i.e., mid-portion of the pelvis) to cover the mid 5-m section of the runway. The following statement was added to the methods section.

Methods: Running speed of participants was calculated based on the time taken for the right hip of participants (i.e., mid-portion of the pelvis) to cover the mid 5-m section of the runway. The following statement was added to the methods section.

The last paragraph of Section Materials and Methods includes an URL citation, which should appear in the list of references.

Response: The citation is now referenced appropriately. We thank the reviewer for noticing.

[1] Souza, R.B. : ''An Evidence-Based Videotaped Running Biomechanics Analysis''. Physical Medicine and Rehabilitation Clinics of North America, 2016, vol.27, no.1, pp217-36. DOI:10.1016/j.pmr.2015.08.006.

[2] Bencsik, L. and Zelei, A.: ''Effects of human running cadence and experimental validation of the bouncing ball model''. Mechanical Systems and Signal Processing, 2016, vol.89, pp78-87. DOI:10.1016/j.ymssp.2016.08.001.

[3] Fellin, R.E. and Manal, K and Davis, I.S.: ''Comparison of Lower Extremity Kinematic Curves During Overground and Treadmill Running''. Journal of Applied Biomechanics, 2010, vol.26, no.4, pp407-414.

Round 2

Reviewer 1 Report

Very happy for the changes made, however, I have a few required additions:

1) Table 2 – rather than a key, detailing inter and intra rater, replace inter rater by rater 1 and rater 2 and intra-rater by rating 1 and rating 2

2) The explanation of how the data was pooled (as given in your response) is helpful and should be included in the text (although you say each participant, but I think you mean rater). However, who rated the intra rater reliability data (both thus 336 trials or just one rater so 168 trials) - please clarify. 

3) You need to report the two-way random effects, absolute agreement, single rater/measurement (2,1) ICC values rather than the 3,1 as these are not meant for wider use and cannot be used by others . 

Author Response

Manuscript ID: sports-394548
Title: Reliability of overground running measures from 2D video analyses in a field environment
Journal: Sports

Response: We appreciate the opportunity to submit a revised version of our manuscript to Sports for publication consideration after minor revision. As requested, we have incorporated revisions according to the comments from your panel of expert reviewers and provide below a point-by-point summary of how we have addressed each comment. We have attempted to be as specific as possible in our responses. In addition, to facilitate the review process, we have indicated all modifications in the manuscript in RED font colour, except for removed text where there is no colour coding. We thank the editors and reviewers of Sports for their rapid turnaround time and critical appraisal of our work.

Reviewer 1

Very happy for the changes made, however, I have a few required additions:

1) Table 2 – rather than a key, detailing inter and intra rater, replace inter rater by rater 1 and rater 2 and intra-rater by rating 1 and rating 2

Response: We have amended Table 2. Rather than using “rating”, we have elected to use “occasion” for consistency with our text and figures.

2) The explanation of how the data was pooled (as given in your response) is helpful and should be included in the text (although you say each participant, but I think you mean rater). However, who rated the intra rater reliability data (both thus 336 trials or just one rater so 168 trials) - please clarify.

Response: An explanation of how the data was pooled is included in the text. We now also specify that both raters contributed to the intra-rater reliability analysis.

Methods: Participants were asked to run three times (…) both pre- and post-race, for a total of 6 running trials for each participant and 168 potentially eligible videos for intra- and inter-rater reliability assessment (28 participants x 2 sessions x 3 trials). Due to the on-field nature of the data collection, 13 of the potentially eligible videos were not available for subsequent reliability assessment (…). Hence, reliability analyses were performed on 155 video recordings.

Methods: Both raters completed the data extraction from the 155 eligible videos twice; therefore, 310 comparisons were involved in determining intra-rater and inter-rater reliability values.

3) You need to report the two-way random effects, absolute agreement, single rater/measurement (2,1) ICC values rather than the 3,1 as these are not meant for wider use and cannot be used by others .

Response: We agree with the reviewer that ICC (3,1) limits the generalisability of findings to other raters, as is highlighted in our limitations section. We believe that using a two-way mixed-effects model rather than a two-way random effects model is more suited to the data collected (i.e., specific raters) and is the most appropriate way to present the true current data set.

Reviewer 3 Report

The Reviewer's concerns have been considered and the suggested corrections have been included in the manuscript.

Author Response

Manuscript ID: sports-394548
Title: Reliability of overground running measures from 2D video analyses in a field environment
Journal: Sports

Response: We appreciate the opportunity to submit a revised version of our manuscript to Sports for publication consideration after minor revision. As requested, we have incorporated revisions according to the comments from your panel of expert reviewers and provide below a point-by-point summary of how we have addressed each comment. We have attempted to be as specific as possible in our responses. In addition, to facilitate the review process, we have indicated all modifications in the manuscript in RED font colour, except for removed text where there is no colour coding. We thank the editors and reviewers of Sports for their rapid turnaround time and critical appraisal of our work.

Reviewer 3
The Reviewer's concerns have been considered and the suggested corrections have been included in the manuscript.

Response: We thank the reviewer for his / her time and contribution to the peer review of our work.